# A minimal mechanistic model of plant responses to oxygen deficit during waterlogging

Silou Chen[1,2], Hugo J. de Boer[1] and Kirsten ten Tusscher[2,3] 

[1]Copernicus Institute of Sustainable Science, Department of Geosciences, Utrecht University, Utrecht, The Netherlands; [2]Theoretical Biology, Department of Biology, Utrecht University, Utrecht, The Netherlands; [3]Experimental and Computational Plant Development, Department of Biology, Utrecht University, Utrecht, The Netherlands

## Original Research Article

**Keywords:**
aerenchyma; flooding; mechanistic modelling; radial oxygen loss barrier; rooting depth.

**Corresponding author:**
Kirsten ten Tusscher;
Email: k.h.w.j.tentusscher@uu.nl

**Associate Editor:**
Prof. Iain Johnston

## Abstract

Plants exhibit diverse morphological, anatomical and physiological responses to hypoxia stress from soil waterlogging, yet coordination between these responses is not fully understood. Here, we present a mechanistic model to simulate how rooting depth, root aerenchyma -porous tissue arising from localized cell death-, and root barriers to radial oxygen loss (ROL) interact to influence waterlogging survival. Our model revealed an interaction between rooting depth and the relative effectiveness of aerenchyma and ROL barriers for prolonging waterlogging survival. As the formation of shallow roots increases waterlogging survival time, the positive effect of aerenchyma becomes more apparent with increased rooting depth. While ROL barriers further increased survival in combination with aerenchyma in deep-rooted plants, ROL barriers had little positive effect in the absence of aerenchyma. Furthermore, as ROL barriers limit root-to-soil oxygen diffusion bidirectionally, our model revealed optimality in the timing of ROL formation. These findings highlight the importance of coordination between morphological and anatomical responses in waterlogging resilience of plants.

## 1. Introduction

Soil waterlogging is a major abiotic stress that constrains plant growth and development. Waterlogging results in waterfilled soil pores, causing a drastic reduction in gas content and diffusion (Armstrong, 1980). As a consequence, the soil becomes anaerobic over the course of a few hours to a few days (Adegoye et al., 2023; Patrick & Delaune, 1977), triggering plants to switch to anaerobic metabolism to ensure energy production (Parent et al., 2008; Sairam et al., 2008). However, according to the Pasteur effect, this requires about 15 times as much glucose as aerobic metabolism. At the same time, root oxygen deficit causes a decline in root hydraulic conductivity due to gating of aquaporins and thereby leads to (partial) stomatal closure and reduction in photosynthesis (Ahmed et al., 2002; Törnroth-Horsefield et al., 2006; Bashar et al., 2019). Combined, this causes plants to become prone to mortality from carbon starvation (Bansal & Srivastava, 2015; Camisón et al., 2020).

Plant shoot–root ratio is a key factor determining plant tolerance against water stress, such as drought and waterlogging (Comas et al., 2013). Mašková et al. (2022) selected 15 genera of plant species for which optimal soil moisture levels ranged from dry to moist habitats. The authors found a positive correlation between length-based shoot–root ratio and soil moisture level, in line with previous findings that deep-rooting plants have better access to soil water under drought (Vanaja et al., 2011; Comas et al., 2013; Maurel & Nacry, 2020). The Mašková et al. (2022) results also imply less allocation to roots in plants adapted to moist habitats, consistent with observations from Fan et al. (2017) showing that waterlogging-adapted plants typically exhibit less rooting depth, presumably to mitigate oxygen stress (Fan et al., 2017).

In addition to the morphological adaptation of a shortened root system, flooding-tolerant plant species display further anatomical adaptations, most notably the presence and further induction of aerenchyma and radial oxygen loss (ROL) barriers (Chen et al., 2023; Colmer, 2003a; Van Der Weele et al., 1996). Formation of aerenchyma involves cell death mediated partial conversion of the parenchyma into an air space, thereby increasing tissue porosity and root oxygenation (Steffens, 2014). Radial oxygen loss (ROL) barriers consist of a suberin-rich structural layer formed around the root endodermis and/or exodermis to prevent radial oxygen

loss to the waterlogged soil (Peralta Ogorek et al., 2023). In nature, most plant species able to form ROL barriers are wetland species (Ejiri et al., 2021), which typically contain constitutive aerenchyma that can be further enhanced during flooding stress (Evans, 2004; Jung et al., 2008). Some upland plant species, seeded during dry seasons and grown in rainfed fields, e.g., maize and wheat, can induce aerenchyma under flooding stress but do not contain constitutive aerenchyma (Pedersen, Sauter, et al., 2021). ROL barriers are usually not observed in these species (Shiono et al., 2011), with a few exceptions such as upland rice and teosinte (Colmer, 2003b; Mano et al., 2006). To the best of our knowledge, ROL barriers have not been observed in plant species that do not form aerenchyma.

The mechanism underlying the co-occurrence of ROL barriers with aerenchyma, while aerenchyma can occur in isolation, has thus far not been investigated. Formation of constitutive aerenchyma results from differential growth, during which some adjacent cells are separated from one another, and air spaces are formed (Evans, 2004). Under waterlogging conditions, ethylene rapidly accumulates and induces additional aerenchyma formation (Bailey-Serres & Voesenek, 2008), whereas ROL barriers are induced downstream of rhizosphere-localized reductive phytotoxins that gradually accumulate as waterlogged soils become anoxic (Shiono et al., 2008, Shiono et al., 2011). Thus, ROL barrier induction does not appear to occur downstream of aerenchyma formation. Instead, both adaptations are related to shoot–root ratio (Lynch et al., 2021; Shiono et al., 2011).

Over the last decades, computational modelling on either the field or single plant level has proven valuable to understand the interplay among plant properties, environmental conditions and plant growth or yield (Beegum et al., 2023; Liu et al., 2020; Shaw et al., 2013). However, to the best of our knowledge thus far models incorporating morphological and anatomical adaptations to waterlogging and their effects on plant physiology and fitness have not been developed. In this study, we set out to build a mathematical model that simulates oxygen dynamics, carbohydrate status and survival in plants with different rooting depths under the presence of different acclimation strategies to decipher the mechanistic basis underlying the relations among rooting depth, aerenchyma content and ROL barrier formation.

## 2. Materials and methods

### 2.1. Plant architecture and plant environment architecture in the model assumptions regarding modelled plant architecture

We modelled plants as consisting of two discrete compartments, the shoot and root, capable of exchanging oxygen and carbon (Figure 1). In our theoretical approach, the shoot consisted of a stem and a canopy of constant size. Here, the stem was represented by a cylinder and the canopy was modelled as a single 'big leaf', an approach frequently used in soil–plant–atmosphere–continuum models that simulate water transport in plants (Damm et al., 2020). We assume photosynthesis only occurs in the 'big leaf', ignoring the minor potential contribution by stems. The root system was represented by a single cylinder with the same diameter as the stem and variable depth to represent different rooting depths ($Z_r$). We thus ignored effects of plant architecture related to, e.g. branching of shoots and roots, as well as the development of this architecture over time as the plant ages. The parameters for the model plant architecture, their default values and experimental value ranges are shown in Supplementary Table S1.

In our simulations, we explored the impact of rooting depth, aerenchyma and ROL barriers (the control parameters in our model) on plant oxygen dynamics, metabolic rate and state and survival time under anoxic conditions (the output of our model). We ignored potential additional effects from changes in cross-sectional root structure, lateral root density, length or angle or the induction of adventitious roots. Oxygen dynamics were modelled on the shoot and root compartment levels. For the air, we assumed a constant partial oxygen pressure of 20 kPa. The rhizosphere and the bulk soil are represented as concentric ring columns surrounding the root cylinder (Figure 1). Given that the rhizosphere is a relatively thin soil layer directly surrounding the root we set the radius of the rhizosphere $R_{rhizo}$ to $2R_r$. In contrast, the bulk soil is the soil surrounding the rhizosphere and, because of its larger volume, is assumed to display more buffered dynamics. To ensure this buffered dynamics, we set the radius of the bulk soil $R_{bulk}$ to $4R_{rhizo}$. The depths of rhizosphere and bulk soil are set as $Z_{rhizo} = Z_r + R_{rhizo}$ and $Z_{bulk} = Z_r + R_{rhizo} + R_{bulk}$, respectively. Therefore, the widths of the rings of rhizosphere and bulk soil columns are $R_r$ and $6R_r$, respectively. $V_{rhizo}$ and $V_{bulk}$ are then calculated as $\pi R_{rhizo}^2 (Z_{rhizo}) - V_{root}$ and $\pi R_{bulk}^2 (Z_{bulk}) - V_{root} - V_{rhizo}$, respectively (Figure 1). Oxygen exchange occurred between the rhizosphere and bulk soil, air and shoot, rhizosphere and root, shoot and root, and air and both soil compartments. Carbohydrate reserve dynamics were modelled on the whole plant level. Exchange of carbon with the surroundings, such as exudation or decay of plant material, was ignored. Waterlogging, i.e., flooding up to but not beyond the soil surface, was simulated through a substantial decrease in air–soil and inter-soil oxygen diffusion rate (for details see below).

### 2.2. Oxygen dynamics

For oxygen dynamics in the shoot and root, we wrote:

$$\frac{d[O_2]_{shoot}}{dt} = (Q_{PO}C_v + Q_{SDO} - Q_{SRO} - Q_{SCO}C_v)/V_{shoot} \quad (1)$$

$$\frac{d[O_2]_{root}}{dt} = (Q_{RDO} + Q_{SRO} - Q_{RCO}C_v)/V_{root} \quad (2)$$

Since air oxygen levels are typically expressed as a partial pressure, we decided to use as units for $[O_2]$ atm (standard atmospheric pressure), with 1 atm$=10^5$Pa . From this, it follows that d$[O2]$/d$t$ is expressed in atm/s and that the (scaled) $O_2$ fluxes $Q_{xx}$ are expressed in atm m$^3$/s. Following the ideal gas law which states that $PV = nRT$, where $P$ is pressure in Pa, $V$ is volume in m$^3$, $n$ is number of molecules in mol, $R$ is the ideal gas constant (8.314 m$^3$ Pa/(K mol)) and $T$ is temperature (294.15 Kelvin for 20 °C), we note that to convert 1 mol/s into atm m$^3$/s we need to multiply with RT$*10^{-5}$ ($10^{-5}$ to take into account that $1(mol/m^3)*RT$ equals 1 Pa, and thus$10^{-5}$ atm). For simplicity, we use the conversion constant $C_v$, with $C_v = $ RT$^*10^{-5}$, in our equations to convert those fluxes that are expressed in mol/s to fluxes in atm m$^3$/s. The various fluxes are due to photosynthetic oxygen production ($Q_{PO}$), oxygen diffusion between air and shoot ($Q_{SDO} > 0$ if oxygen flows from air into the shoot), oxygen exchange between the shoot and root ($Q_{SRO}$) ($Q_{SRO} > 0$ if oxygen flows from shoot to root), shoot respiratory oxygen consumption ($Q_{SCO}$), root rhizosphere oxygen exchange ($Q_{RDO}$) ($Q_{RDO} > 0$ if oxygen flows from the rhizosphere to the root) and finally, root respiratory oxygen consumption ($Q_{RCO}$).

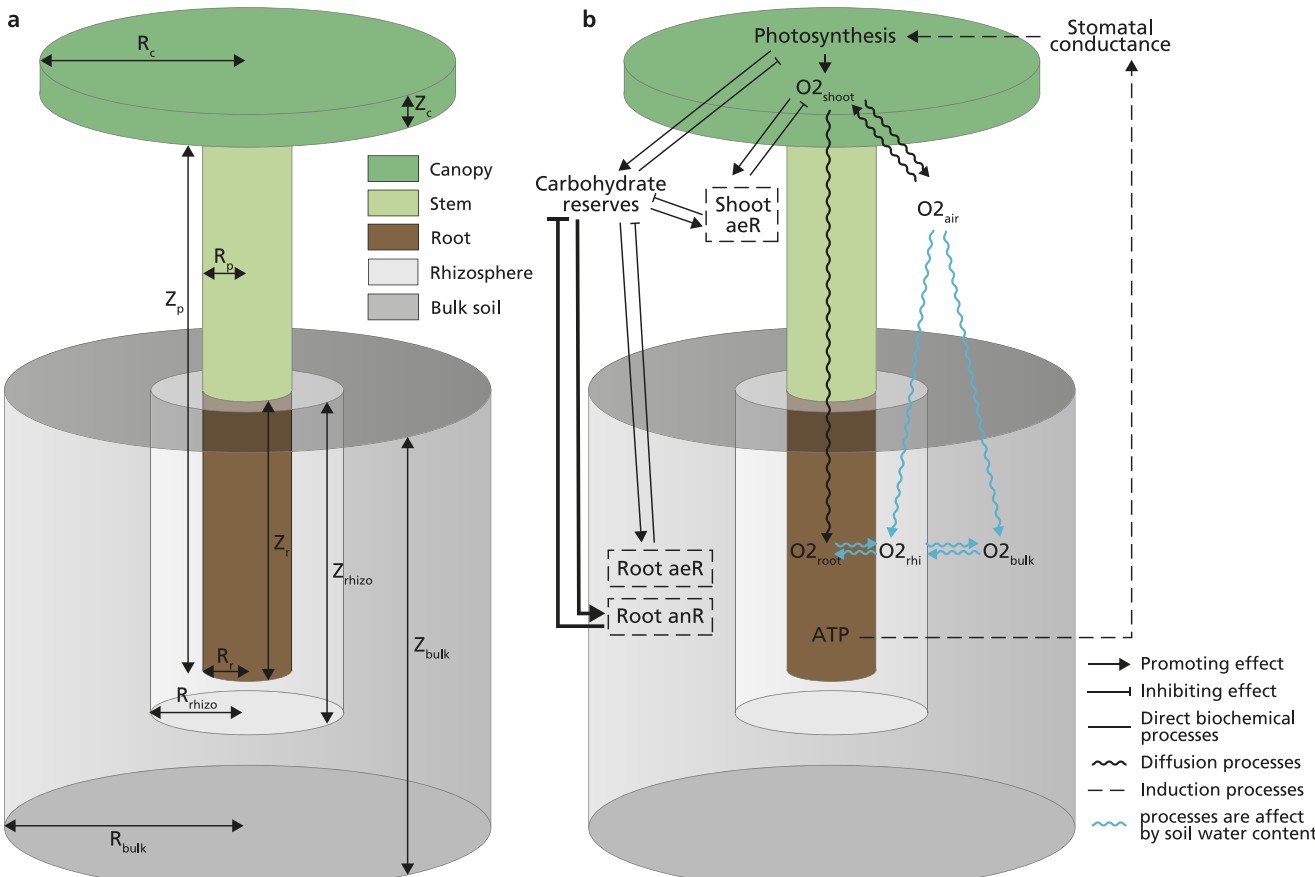

**Figure 1.** Overview of the model layout. (a) The modelled plant consists of one round "big leaf" as the canopy, a stem, which together with the canopy makes the shoot, and a root. As the environment, we consider the rhizosphere directly surrounding the root, a limited volume of bulk soil, and the atmosphere surrounding the plant shoot. (b) The model simulates the exchange of oxygen between atmosphere and shoot, shoot and root, atmosphere and bulk and rhizosphere soil, bulk soil and rhizosphere and rhizosphere and root. Of these, the latter 4 are significantly reduced under waterlogging conditions. The model simulates how photosynthesis-mediated carbon synthesis and the usage of carbohydrates in aerobic and anaerobic respiration and concomitant ATP production depend on shoot and root oxygen levels, with root ATP levels feeding back on stomatal aperture and hence photosynthesis. Aerenchyma presence enhances shoot root oxygen exchange, while ROL barrier presence reduces rhizosphere root oxygen exchange.

To convert flows into concentrations, they are divided by shoot volume $(\pi R_c^2 Z_c + \pi R_p^2 Z_p)$ and root volume $(\pi R_r^2 Z_r.)$.

Given the chemical equation for photosynthesis:

$$6CO_2 + 12H_2O + h\nu \xrightarrow[enzymes]{chlorophyll} C_6H_{12}O_6 \,(glucose) + 6H_2O + 6O_2 + ATP$$

Photosynthetic oxygen production rate is six times that of carbohydrate assimilation. Therefore, we wrote. $Q_{PO} = 6Q_{PC}$, where $Q_{PC}$ denotes the flow of carbohydrate assimilation through photosynthesis. To keep our model as simple as possible, we ignored the effects of light quality and quantity, $CO_2$ vapour pressure deficit and temperature on photosynthesis frequently incorporated in other models, and here, instead only incorporated the effect of waterlogging. Of course, if we were to investigate how combining waterlogging with low light or high temperature stresses aggravate the risk of carbon starvation, these factors would also need to be taken into consideration. Waterlogging leads to root hypoxia and thus energy exhaustion, which, due to the accumulation of lactate and lack of ATP to drive proton ATPases, causes root acidification and subsequent aquaporin gating (Kudoyarova et al., 2022; Tournaire-Roux et al., 2003). As a result, root water uptake is reduced, plant water potential drops and stomata are (partially) closed, reducing the photosynthetic rate. It is this process that causes plant waterlogging responses to partially overlap with plant drought responses.

To keep our model simple, we refrained from a full modelling of root metabolism, pH, aquaporin gating and plant hydraulics that would be necessary to mechanistically link flooding to stomatal aperture changes. Instead, we used plant root ATP status as a proxy to control stomatal aperture, and we assume photosynthetic rate is linearly controlled by the stomatal aperture. We also incorporated feedback inhibition of carbohydrate levels on photosynthetic rate (Paul & Foyer, 2001; Rosado-Souza et al., 2023). We thus simulated these processes using the following formula:

$$Q_{PO} = A_{O_2} g S_{canopy} \tag{3}$$

$$A_{O_2} = A_{max} \frac{K_A^n}{K_A^n + [C_6H_{12}O_6]^n} \tag{4}$$

$$g = \frac{ATP_{root}^m}{ATP_{root}^m + K_{ATP}^m}(1 - \beta_g) + \beta_g \tag{5}$$

With $g$ the fractional stomatal aperture, $A_{O_2}$ the maximum photosynthesis rate remaining from negative feedback carbohydrate inhibition, $A_{max}$ denotes the maximum rate of photosynthetic oxygen production, $K_A$ the carbohydrate level that leads to a half maximum photosynthetic rate, $K_{ATP}$ the ATP concentration leading to half maximum stomatal aperture and $\beta_g$ the baseline stomatal aperture sustained during waterlogging (Supplementary Table S1)

and $S_{\text{canopy}}$ the area of the single big leaf canopy, calculated as $\pi R_c{}^2$.

The oxygen exchange processes for the plant were modelled as:

$$Q_{\text{SDO}} = D_{\text{shoot}}g \frac{\left([O_2]_{\text{air}} - [O_2]_{\text{shoot}}\right)}{Z_c/2} \cdot S_{\text{shoot}} \tag{6}$$

$$Q_{\text{SRO}} = \frac{D_{\text{plant}}\left(1 + \alpha_{AeT}[AeT]\right)\left([O_2]_{\text{shoot}} - [O_2]_{\text{root}}\right)}{(Z_{\text{p}} + Z_{\text{r}})/2} \cdot S_{\text{cross}} \tag{7}$$

$$Q_{\text{RDO}} = D_{\text{root}} \frac{\left([O_2]_{\text{rhizo}} - [O_2]_{\text{root}}\right)}{1.5R_r} \left(1 - [ROLB]\right) \cdot S_{\text{root}} \tag{8}$$

with $D_{\text{shoot}}$ the rate of effective air–shoot diffusion through fully opened stomata is taken equal to $D_{\text{air}}$. Note that we thus simplify air–shoot oxygen exchange as a diffusive process, while in reality, vapour pressure deficit-driven conductive efflux that may even act against a stomatal–air oxygen gradient is likely playing a major role. Shoot area $S_{\text{shoot}} = 2\pi R_{\text{p}} Z_{\text{p}} + S_{\text{canopy}}$, $D_{\text{plant}}$ the baseline effective shoot-root diffusion rate, $[AeT]$ the cross-sectional aerenchyma fraction, $\alpha_{AeT}$ the maximum increase in shoot-root diffusion rate if $[AeT]$ approaches 1, $S_{\text{cross}} = 2\pi R_r$ the cross-sectional area connecting shoot and root, and $(Z_{\text{p}} + Z_{\text{r}})/2$ scaling the diffusion rate with shoot–root distance, $D_{\text{root}}$ the maximum soil–root oxygen conductance, and $[ROLB]$ the fractional reduction of effective diffusion due to ROL barrier formation, with $[ROLB]$ having a maximum value of 0.9 to consider the absence of ROL barrier formation at the root cap, and root surface area $S_{\text{root}} = \pi R_r{}^2 + 2\pi R_r Z_r$. Rhizosphere root oxygen exchange is scaled with the distance between the root cylinder and the rhizosphere soil ring ($1.5R_r$) Assuming that in the path from soil to root, diffusion in soil is the limiting factor, we take $D_{\text{root}} = D_{\text{soil}}$, for which we use

$$D_{\text{soil}} = D_{\text{air}} \frac{\theta^{\frac{10}{3}}}{f^2} \frac{K_{\text{root}}^z}{H^z + K_{\text{root}}^z} + D_{\text{water}} \frac{H^z}{H^z + K_{\text{root}}^z} \tag{9}$$

With $D_{\text{air}}$ the diffusion rate of oxygen in air, $\varnothing$ the fraction of gas-filled soil pores, $f$ the soil porosity fraction (Jin and Jury, 1996), $H$ the soil water level (height in m), and $K_{\text{root}}$ the soil water content (height) at which the effective oxygen diffusion coefficient in the soil reaches $\frac{D_{\text{water}} + D_{\text{air}}}{2}$. To simulate the drastic decrease in oxygen diffusion if soil pores are fully water-filled we used $z = 10$, to ensure a sudden transition.

For shoot and root respiratory oxygen consumption, we assumed a saturating dependence on oxygen concentration and a simple linear dependence on glucose level (Päpke et al., 2014).

$$Q_{\text{SCO}} = \left( \frac{m_{O_2\text{shoot}}[O_2]_{\text{shoot}}^p}{[O_2]_{\text{shoot}}^p + h_{O_2\text{shoot}}^p} \beta_m + \frac{m_{O_2\text{shoot}}[O_2]_{\text{shoot}}^p}{[O_2]_{\text{shoot}}^p + h_{O_2\text{shoot}}^p} \right.$$
$$\left. [C_6H_{12}O_6]\, m_g\,(1 - \beta_m) \right) \cdot W_{\text{shoot}} \tag{10}$$

$$Q_{\text{RCO}} = \left( \frac{m_{O_2\text{root}}[O_2]_{\text{root}}^p}{[O_2]_{\text{root}}^p + h_{O_2\text{root}}^p} \beta_m + \frac{m_{O_2\text{root}}[O_2]_{\text{root}}^p}{[O_2]_{\text{root}}^p + h_{O_2\text{root}}^p} \right.$$
$$\left. [C_6H_{12}O_6]\, m_g\,(1 - \beta_m) \right) \cdot W_{\text{root}} \tag{11}$$

With $m_{O_2\text{shoot}}$ and $m_{O_2\text{root}}$ the maximum shoot and root oxygen consumption rate, $h_{O_2\text{shoot}}$ and $h_{O_2\text{root}}$ the shoot and root oxygen concentration at which the oxygen consumption rate is half-maximal, $m_g$ the molecular weight of glucose, $\beta_m$ the fraction of carbohydrate-independent respiration, since carbohydrates serve as the main but not only substrate and $W_{\text{shoot}}$ and $W_{\text{root}}$ shoot and root dry weight.

For simplicity, we did not explicitly describe the dynamics of the soil reductive phytotoxins that induce ROL barrier formation. Instead, since soil phytotoxins are formed under rhizosphere oxygen deficit, we took the latter as a proxy for ROL barrier induction:

$$\frac{d[\text{ROLB}]}{dt} = \frac{\alpha_{\text{ROLB}} \max\left([O_2]_{\text{rhizo}}' - [O_2]_{\text{rhizo}}, 0\right)^q}{\max\left([O_2]_{\text{rhizo}}' - [O_2]_{\text{rhizo}}, 0\right)^q + K_{\text{ROLB}}^q} \tag{12}$$

with $\alpha_{\text{ROLB}}$ the rate of ROL barrier formation, $[O_2]_{\text{rhizo}}'$ the threshold of rhizosphere oxygen level below which ROL barriers start to be induced, and $K_{ROLB}$ the rhizosphere oxygen deficit that leads to a half-maximal ROL barrier induction rate. If rhizosphere oxygen level exceeds the threshold value, $\frac{d[\text{ROLB}]}{dt} = 0$. Since Eq. (11) does not contain a decay term, under low oxygen levels, ROLB levels increase at a constant rather than gradually decreasing speed. To prevent ROLB levels from increasing indefinitely, we apply a maximum ROLB level, which under default conditions equals 0.9.

Oxygen dynamics in the rhizosphere and bulk soil were modelled as follows:

$$\frac{d[O_2]_{\text{rhizo}}}{dt} = (Q_{\text{ARhO}} - Q_{\text{RDO}} - Q_{\text{RhBO}})/V_{\text{rhizo}} - Q_{\text{RhCO}} \tag{13}$$

$$\frac{d[O_2]_{\text{bulk}}}{dt} = (Q_{\text{ABO}} + Q_{\text{RhBO}})/V_{\text{bulk}} - Q_{BCO} \tag{14}$$

$Q_{\text{ARhO}}$ and $Q_{\text{ABO}}$ denote the oxygen exchange between air and rhizosphere and bulk soil, respectively ($Q_{\text{ARhO}}, Q_{\text{ABO}} > 0$ if oxygen flows from air into soil), $Q_{\text{RhBO}}$ denotes the oxygen flow between rhizosphere and bulk soil ($Q_{\text{RhBO}} > 0$ if oxygen flows from rhizosphere into bulk soil), $Q_{\text{RhCO}}$ and $Q_{\text{BCO}}$ the oxygen consumption from aerobic respiration in rhizosphere and bulk soil, respectively, and $V_{\text{rhizo}}$ and $V_{\text{bulk}}$ the volume of rhizosphere and bulk soil, respectively.

Assuming a homogeneous soil and air-soil interface, we modelled oxygen exchange between air and soil as:

$$Q_{\text{ARhO}} = D_{\text{soil}} \frac{[O_2]_{\text{air}} - [O_2]_{\text{rhizo}}}{Z_{\text{rhizo}}/2} \cdot S_{\text{rhizo}} \tag{15}$$

$$Q_{\text{ABO}} = D_{\text{soil}} \frac{[O_2]_{\text{air}} - [O_2]_{\text{bulk}}}{Z_{\text{bulk}}/2} \cdot S_{\text{bulk}} \tag{16}$$

$$Q_{\text{RhBO}} = D_{\text{soil}} \frac{[O_2]_{\text{rhizo}} - [O_2]_{\text{bulk}}}{4R_r} \cdot S_{\text{interface}} \tag{17}$$

Note that we assume that for diffusion of oxygen from the air into the soil, diffusion in the soil is limiting and hence we have taken $D_{\text{soil}}$ both for the rate of diffusion from air to soil as for intra soil diffusion. $S_{\text{rhizo}}$ and $S_{\text{bulk}}$ represent the contact areas of rhizosphere and bulk soil with air, calculated as $\pi\left(R_{\text{rhizo}}^2 - R_{\text{root}}^2\right)$ and $\pi\left(R_{\text{bulk}}^2 - R_{\text{rhizo}}^2\right)$, respectively, and $S_{\text{interface}}$ the area of the rhizosphere–bulk soil interface, calculated as $\pi R_{\text{rhizo}}^2 + 2\pi R_{\text{rhizo}}(Z_r + R_r)$. We scaled all three diffusion processes with distance, taking half the heights of the rhizosphere cylinder and bulk soil cylinder as distance for the air to rhizosphere ($Z_{\text{rhizo}}/2$) and air to bulk soil diffusion ($Z_{\text{bulk}}/2$) and taking the distance between the middle of the rhizosphere and bulk soil rings as a distance ($4R_r$).

Assuming homogeneous, identical microorganism distributions across the rhizosphere and bulk soil oxygen consumption from aerobic respiration was modelled as:

$$Q_{\text{RhCO}} = m_{\text{rhizo}} \frac{[O_2]_{\text{rhizo}}^r}{[O_2]_{\text{rhizo}}^r + h_{\text{rhizo}}^r} \cdot V_{\text{rhizo}} \tag{18}$$

$$Q_{\text{BCO}} = m_{\text{bulk}} \frac{[O_2]_{\text{bulk}}^r}{[O_2]_{\text{bulk}}^r + h_{\text{bulk}}^r} \cdot V_{\text{bulk}} \quad (19)$$

$m_{\text{rhizo}}$ and $m_{\text{bulk}}$ denote the maximum oxygen consumption rates in the rhizosphere and bulk soil. $h_{\text{rhizo}}$ and $h_{\text{bulk}}$ denote the oxygen concentration in rhizosphere and bulk soil at which the oxygen consumption rate is half-maximal.

## 2.3. Carbohydrate reserve dynamics

Carbohydrate reserve dynamics were modelled on a whole plant level, taking into consideration that carbohydrates are only produced in the shoot while their consumption occurs in both root and shoot, and carbohydrate reserves are consumed through respiration, given ample oxygen, or through fermentation, given oxygen deficiency. Therefore, we wrote:

$$\frac{d[C_6H_{12}O_6]}{dt} = [Q_{\text{PC}} - (Q_{\text{SCC}} + Q_{\text{RCC}})] / (W_{\text{shoot}} + W_{\text{root}}) \quad (20)$$

where $[C_6H_{12}O_6]$ is expressed in mol per gram plant dry weight, and hence fluxes are in mol/s. $Q_{\text{PC}}$ is the rate of photosynthesis, equal to $\frac{Q_{\text{PO}}}{6}$, where $Q_{\text{PO}}$ is our previously formulated production rate of oxygen through photosynthesis and $Q_{\text{SCC}}$ and $Q_{\text{RCC}}$ denote shoot carbohydrate and root carbohydrate consumption rate (mol s$^{-1}$), and the dry mass of the whole plant $W_{\text{shoot}} + W_{\text{root}}$ serves to translate carbohydrate amounts into dry weight fractions. Our approach thus ignores details of shoot-to-root carbon allocation and possible changes therein under flooding.

Shoot and root carbohydrate consumption ($Q_{\text{SCC}}$ and $Q_{\text{RCC}}$, respectively) each consists of aerobic and anaerobic consumption, calculated as.

$$Q_{\text{SCC}} = Q_{\text{SAC}} + Q_{\text{SNC}} \quad (21)$$

$$Q_{\text{RCC}} = Q_{\text{RAC}} + Q_{RNC} \quad (22)$$

$Q_{\text{SAC}}$ and $Q_{\text{SNC}}$ denote the carbohydrate fluxes due to shoot aerobic and anaerobic consumption, $Q_{\text{RAC}}$ and $Q_{\text{RNC}}$ denote the carbohydrate fluxes due to root aerobic and anaerobic consumption.

Given the chemical equation for glucose metabolism:

$$C_6H_{12}O_6 \text{ (glucose)} + 6O_2 \xrightarrow{\text{enzymes}} 6CO_2 + 6H_2O + 36ATP$$

We expressed the aerobic carbohydrate consumption of the shoot and root as a function of oxygen consumption:

$$Q_{\text{SAC}} = Q_{\text{SCO}}/6 \quad (23)$$

$$Q_{\text{RAC}} = Q_{\text{RCO}}/6 \quad (24)$$

Anaerobic carbohydrate consumption results from glycolysis and fermentation. According to the Pasteur effect, anaerobic metabolism requires 18 times more carbohydrate input for the same amount of energy production compared to aerobic metabolism. Therefore, we wrote anaerobic carbohydrate consumption as:

$$Q_{\text{SNC}} = \left( \frac{\frac{18 m_{O_2\text{shoot}}}{6} h_{O_2\text{shoot}}^p}{[O_2]_{\text{shoot}}^p + h_{O_2\text{shoot}}^p} \beta_{\text{m}} + \frac{\frac{18 m_{O_2\text{shoot}}}{6} h_{O_2\text{shoot}}^p}{[O_2]_{\text{shoot}}^p + h_{O_2\text{shoot}}^p} \right.$$
$$\left. [C_6H_{12}O_6] m_{\text{g}} (1 - \beta_{\text{m}}) \right) \cdot W_{\text{shoot}} \quad (25)$$

$$Q_{\text{RNC}} = \left( \frac{\frac{18 m_{O_2\text{root}}}{6} h_{O_2\text{root}}^p}{[O_2]_{\text{root}}^p + h_{O_2\text{root}}^p} \beta_{\text{m}} + \frac{\frac{18 m_{O_2\text{root}}}{6} h_{O_2\text{root}}^p}{[O_2]_{\text{root}}^p + h_{O_2\text{root}}^p} \right.$$
$$\left. [C_6H_{12}O_6] m_{\text{g}} (1 - \beta_{\text{m}}) \right) \cdot W_{\text{root}} \quad (26)$$

where the division by 6 converts oxygen production rate to the rate of aerobic carbohydrate metabolism and the multiplication by 18 converts this to the rate of anaerobic carbohydrate consumption.

## 2.4. Root ATP dynamics

ATP dynamics, used to control stomatal aperture (mechanisms explained in the section ***Oxygen dynamics***) was only modelled for the root. ATP is produced during both aerobic and anaerobic respiration, generating 36 or 2 molecules of ATP per molecule of glucose, respectively (Lloyd et al., 1983). We assume ATP consumption to be proportional to concentration. Therefore, we wrote:

$$\frac{d[\text{ATP}]_{\text{root}}}{dt} = (36Q_{\text{RAC}} + 2Q_{\text{RNC}}) / V_{\text{root}} - \delta[\text{ATP}]_{\text{root}} \quad (27)$$

where $[\text{ATP}]_{\text{root}}$ is expressed in mol m$^{-3}$ s$^{-1}$, and with $Q_{\text{RAC}}$ and $Q_{\text{RNC}}$ root aerobic and anaerobic carbohydrate consumption rate, respectively, $\delta$ root ATP consumption rate and $V_{\text{root}}$ root volume, used to convert root ATP molecule numbers into concentration.

Given that the dynamics of ATP is rapid, we assumed that it is a quasi-equilibrium process, allowing us to use:

$$[ATP]_{\text{root}} = (36Q_{\text{RAC}} + 2Q_{\text{RNC}}) / (V_{\text{root}} \delta) \quad (28)$$

## 2.5. Simulation design for waterlogging

We focused on the dynamics of root oxygen concentration, stomatal aperture and carbohydrate reserves to represent plant fitness and survival. We first run the model for 20 days under non-stressed conditions to reach steady state, after which waterlogging is introduced for 20 days (480 hours). We assume plant death when carbohydrate reserves drop to 10% of the initial level, then the simulation stops. We ignore the diurnal cycle, with daytime photosynthesis and nighttime starch mobilization, which would require a more complex modelling of carbohydrate dynamics and instead assume constant light and photosynthesis.

As a validation process, we first parametrized our plant architecture based on soybean plants at R1 stage (parameter values shown in Supplementary Table S2), and compared our model output with experimental data from Adegoye et al. (2023). We extracted soil oxygen concentrations (Figure 1 from Adegoye et al. (2023)) and stomatal conductance (Figure 2c from Adegoye et al. (2023)) during waterlogging. To convert stomatal conductance to aperture, we normalized by the maximum conductance under control conditions.

In our simulations, we varied rooting depth, aerenchyma content and ROL barrier levels, while keeping root and shoot diameter, canopy size, shoot length and shoot dry weight constant. Root dry weight was set linearly proportional to the rooting depth, with R1 stage soybean root dry weight as reference (Supplementary Table S2). To enable a fair comparison of survival versus carbohydrate starvation as a function of rooting depth effects on oxygen levels, without reduced rooting depth contributing to enhanced survival due to a relatively larger photosynthetic potential relative to overall plant size, we normalized maximum photosynthetic rates

according to the shoot and root dry weights:

$$A_{max} = A_{ref} \frac{W_{shoot} + W_{root}}{W_{shoot} + W_{ref}} \qquad (29)$$

We prescribed static aerenchyma content levels while dynamically modeling the induction of ROL barriers, following observations that in species inducing ROL barriers, aerenchyma are constitutively present.

## 3. Results

### 3.1. Model validation

In our first simulation we validated our model by using soybean as well as soil type-specific parameter settings, comparing model outcomes to the experimental data from Adegoye et al. (2023). Our model was well able to reproduce the drastic decline in soil oxygen concentration (Supplementary Figure S1a), yet model rhizosphere oxygen levels did not decline to zero as in the experimental data during long-term waterlogging. This discrepancy may arise from the fact that in our model, the degradation of the rhizosphere oxygen is oxygen concentration dependent, the decline slows as levels become lower. For stomatal aperture, our simulation result showed an approximate 10-hour delay in the initial decline relative to experimental data, yet converged to similar levels as in experiments at later time points (Supplementary Figure S1b). We conclude that our simplified model can reasonably faithfully simulate oxygen and stomatal dynamics under waterlogging in a species without ROL barrier and with limited aerenchyma content.

### 3.2. Increased aerenchyma and ROL barrier content promote survival at larger rooting depth

Next, we first set out to obtain a comprehensive overview of the combinations of rooting depths, aerenchyma levels and presence or absence of ROL barrier formation that enable waterlogging survival for otherwise default parameter settings. To investigate a wider range of rooting depths we adjusted soil parameters (see Supplementary Table S1), transitioning from a loamy soil in which even in absence of water logging plants with roots deeper than 0.70 m experience severe anoxia to a sandy soil in which plant roots remain anoxic up to at least 0.90 m of rooting depth. Note that rooting depth affects root volume and hence overall plant metabolic demand, root surface area and hence the soil–root oxygen exchange interface, and shoot–root distance and air–soil–root distance and hence the efficiency of root oxygen delivery.

We found that in the absence of ROL barriers, increasing aerenchyma content up to 0.6–0.7 would enhance rooting depths with which plants can survive up to 0.5 m (Figure 2a). A similar qualitative effect was observed in the presence of maximum ROL barrier induction, yet similar aerenchyma content now allowed larger rooting depths to survive (up to 0.8 m), while for similar rooting depth (0.5 m) to survive less aerenchyma content (0.4 instead of 0.6–0.7) was required (Figure 2b). Our findings suggest that while aerenchyma in isolation promotes waterlogging survival, this effect is enhanced by ROL barriers. Therefore, for plants with deep roots to survive, a significant aerenchyma content combined with ROL barrier formation is essential.

Next, we explored the impact of different ROL barrier levels for a constant level of aerenchyma content (0.5). As before, we observed that in the absence of ROL barriers this aerenchyma level enabled plants with roots no deeper than 0.4 m to survive (Figure 2c). As for aerenchyma, increasing ROL barrier level enhanced the rooting depths for which plants can survive waterlogging, although the effect is less pronounced (from 0.4 to 0.6m rooting depth by increasing to 0.9 ROL barrier level). Since our results suggest that combining ROL barriers with aerenchyma enhances rooting depths that can survive waterlogging and high ROL barrier levels reduce required aerenchyma levels, we wondered to what extent a decrease in one adaptation may be compensated by an increase in the other. To investigate this, we subjected plants with a constant rooting depth of 0.6 m to various combinations of aerenchyma and ROL barrier contents. We found limited compensatory effects, with plants with such rooting depth requiring either a semi-high level aerenchyma content (>0.4) with nearly complete induction of ROL barrier (0.9), or an extremely high level aerenchyma content (>0.7) with low to intermediate level ROL barrier induction (>0.1) (Figure 2d). We concluded that reducing rooting depth, increasing aerenchyma and ROL barrier content can all improve the chance of plant survival during prolonged waterlogging, and they can partly compensate for one another.

### 3.3. Sudden transitions from carbon starvation to long-term survival result from feedback interactions

In addition to showing the importance of rooting depth, aerenchyma and ROLB on plant survival, the results in Figure 2 indicate that changes in rooting depth, aerenchyma and ROLB content result in relatively sudden transitions between short-term survival and subsequent carbon starvation and long-term flooding survival. An important question is to what extent these sudden transitions are a general result of the feedback interactions incorporated in our model (Figure 1b), or rather arise from specific assumed non-linearities in the model implementation of these interactions. To investigate this we varied the two most-critical and strongest non-linearities in our model, the power "p" describing the non-linearity of the dependence of aerobic and anaerobic metabolism on oxygen levels (Eqs. (10)–(11) and (25)–(26)) and the power "m" describing the non-linearity of the dependence of stomatal fractional aperture on ATP levels (Eq. 5). Our results indicate that shallower dependencies for either or both relations (reducing the power of Hill functions from 8 to 2), while shifting the position of the transitions, do not change the sudden nature of these transitions (Supplementary Figure S2), indicating these are inherent to the modelled interactions.

In an effort to define which model interactions underlie the relative suddenness of the non-survival to survival transition in terms of root characteristics, we could identify only a single interaction to have a significant effect on this. Removing the well-known feedback inhibition of carbohydrate levels on photosynthetic rate (Eq. 4) resulted in a transition from non-survival to long-term-survival with more gradually increasing survival times (Supplementary Figure S3). This can be understood from the fact that if, under non-flooding conditions, there is feedback inhibition of photosynthesis, as flooding occurs and carbohydrate reserves drop, this inhibition is alleviated, thereby partly compensating for the carbohydrate decline, thereby offering a certain buffering capacity. While extending the survival parameter regime, beyond the buffering realm, a more sudden transition to short survival times occurs.

### 3.4. Less deep rooting enhances waterlogging survival time

To better understand how the feedbacks in our model lead to survival/non-survival transitions, we investigate different aspects

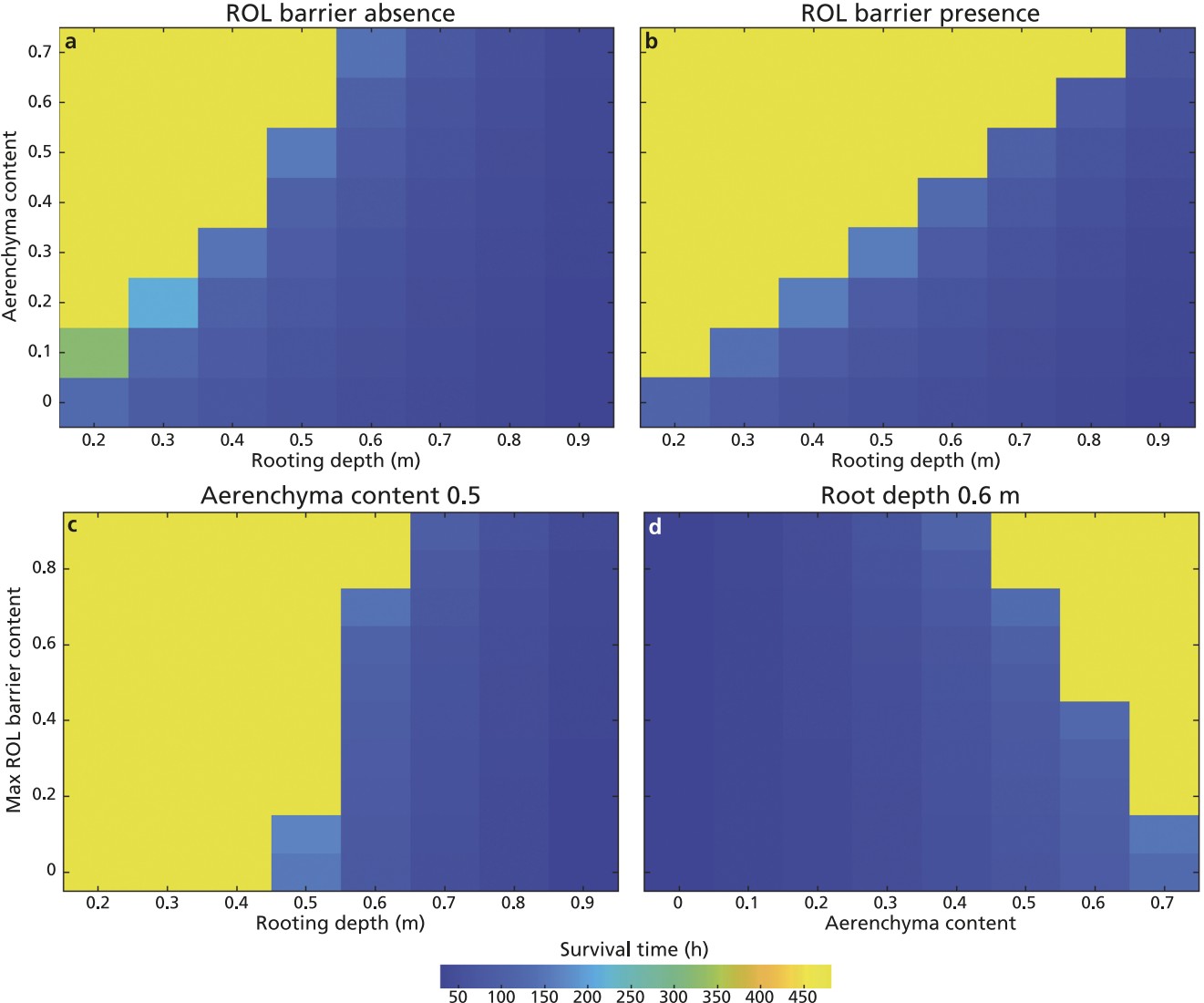

**Figure 2.** The variation in survival time (measured in hours) observed during a 20-day waterlogging treatment across different levels of rooting depths, aerenchyma content levels, and maximum ROL barrier content upon completion. Panel (a) depicts conditions without ROL barriers. Panel (b) showcased conditions with a constant maximum ROL barrier content level at 0.9. Panel (c) maintained a constant aerenchyma content level of 0.5, while panel (d) maintained a constant rooting depth of 0.6 m.

of our model in more detail. First, we investigated the isolated effect of rooting depth on waterlogging survival. Our model showed that upon the initiation of waterlogging, root oxygen levels exhibited a very rapid decline, a subsequent much more gradually declining phase of approximately 10–50 hours depending on rooting depth after which a further decline occurred for all three tested rooting depths of 0.3 m, 0.6 m and 0.8 m (Figure 3a). The rapid decline arises from the instantaneous decline in soil oxygen when pores become waterfilled, limited shoot–root gas diffusivity in the absence of aerenchyma and the high rate of oxygen consumption via aerobic respiration (Supplementary Figure S4a). During the rapid oxygen decline, the root metabolism underwent a shift from predominantly aerobic pathways to predominantly anaerobic pathways (Figure 3b), resulting in a slowdown and eventual stabilization of oxygen consumption. This, however, led to an increased consumption of carbohydrates, causing a decline in carbohydrate levels (Figure 3c), with ATP levels and stomatal aperture that depend on these showing a parallel decline (Supplementary Figure S4b and S4c). Although metabolic rate depends on carbohy-

drate level, causing a slowdown in the decline of carbohydrate levels as it decreases, a constant minimum level of metabolic activity causes the eventual depletion of carbohydrate reserves. Finally, when rhizosphere oxygen level drops below root oxygen level (Supplementary Figure S4d), root oxygen is lost to the rhizosphere, contributing to a secondary decline of root oxygen.

Focusing on the effect of rooting depth, we observed that in non-flooded conditions, a deeper rooting depth results in lower root oxygen levels. This result can be understood from the larger air-to-soil-to-root as well as shoot-to-root distance resulting from larger rooting depths, with in absence of aerenchyma, the former being the major route for oxygen delivery to the root. Under flooding stress, less deep rooting substantially prolonged survival time (i.e. time until depletion of carbohydrate reserves, Figure 3c and d). Specifically, plants with a rooting depth of 0.3 m exhibited a 40 h longer survival than those with a rooting depth of 0.6 m, and the latter survived approximately 10 hours longer than plants with a rooting depth of 0.8 m. These results can be understood from the fact that a shorter air-to-soil-to-root and shoot–root distance

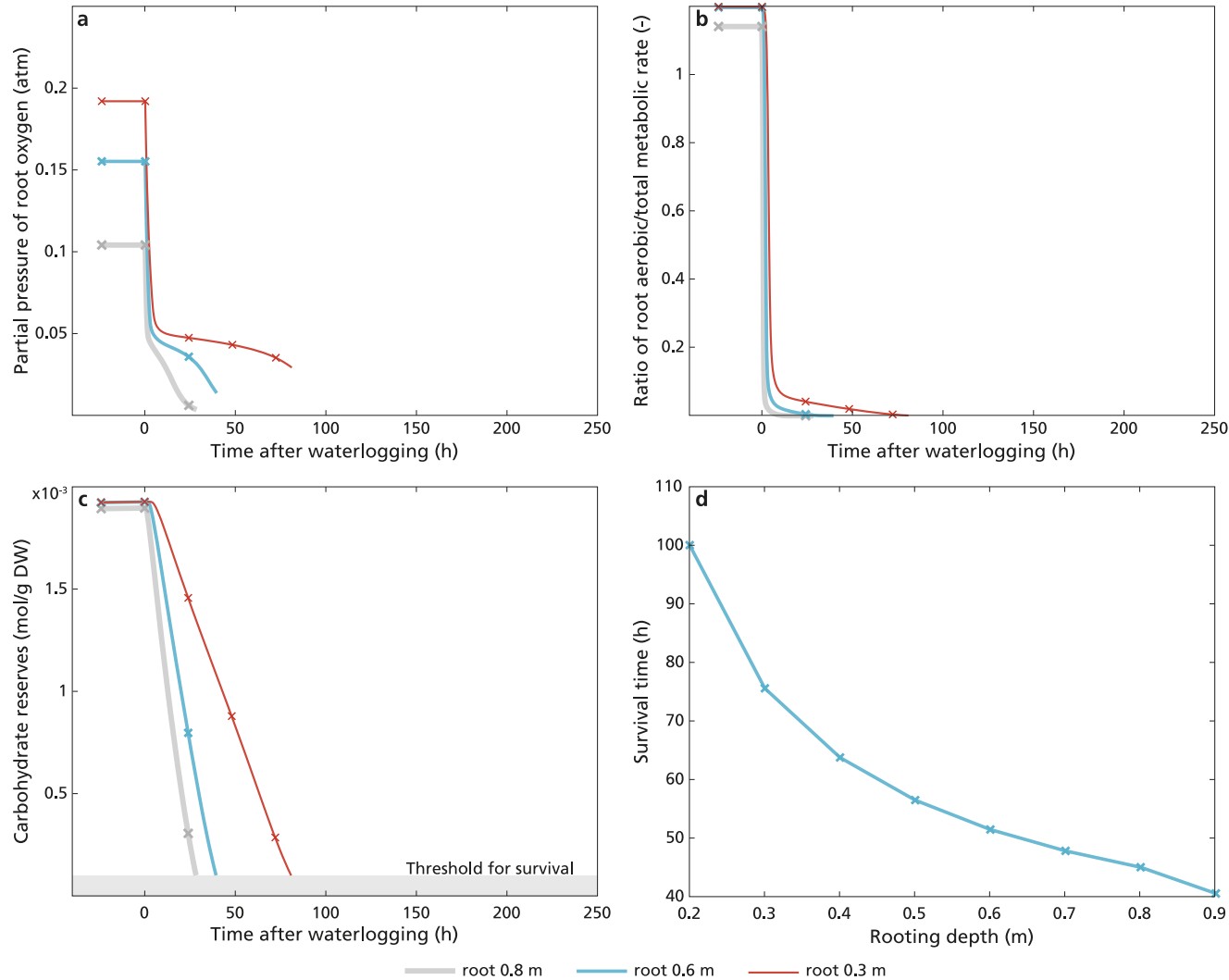

**Figure 3.** Dynamics of (a) root oxygen concentration, (b) the ratio of aerobic metabolic rate in the total metabolic rate, and (c) carbohydrate reserves (plant death occurred when carbohydrate reserves dipped below $10^{-4}$ mol g$^{-1}$ DW, illustrated in grey area) of plants with rooting depths of 0.3 m, 0.6 m, and 0.8 m in the absence of aerenchyma and ROL barriers after 20 days (480 hours) upon the initiation of waterlogging. In panel (d), we presented the survival duration of plants across a range of rooting depths from 0.2 to 0.9, in the absence of aerenchyma during prolonged waterlogging.

enables a more efficient delivery of oxygen to the root and hence a less large decline in root oxygen levels. Consequently, a slightly less pronounced shift to anaerobic metabolism takes place, causing a less rapid decline in carbohydrate reserves, ATP and stomatal aperture, which through the positive relation between stomatal aperture and carbohydrate production further slows down these declines. Nevertheless, our results showed that regardless of the rooting depth, plants were unable to withstand prolonged waterlogging in the absence of additional adaptations.

### 3.5. ROL barriers allow persistent survival for intermediate rooting depths

To further investigate how ROL barriers and aerenchyma may enhance waterlogging survival, we investigated dynamic ROL barrier induction in plants with a 50% cross-sectional aerenchyma content, again varying rooting depth. Specifically, instead of using a value of zero for ROLB in Eq. (8), resulting in unhindered exchange of oxygen between root and soil, we now let the value of ROLB dynamically evolve according to Eq. (11), while using a maximum ROLB value of 0.9. For deep-rooting plants ROL barrier induction

was sped up by 35–40 hours relative to shallower rooted plants (Figure 4a, in-set). This effect can be understood from the fact that greater rooting depth lowers root oxygen and thereby rhizosphere oxygen levels faster, causing a faster induction of ROL barrier formation. Indeed, in experiments deep deep-rooting rice was found to induce and complete ROL barriers quicker than shallow rooting rice, showing similar-sized timing differences (Shiono et al., 2011).

Similar to plants lacking aerenchyma and ROL barriers, waterlogging resulted in a rapid initial decline of root oxygen levels (compare Figures 4a and 3a) under all conditions. However, in the presence of aerenchyma, rooting depth has a much larger effect on the decline in root oxygen during the first 50 hours after the onset of waterlogging. These differences can be attributed to the fact that in absence of aerenchyma, the root hardly receives oxygen from the shoot irrespective of rooting depth, whereas with aerenchyma rooting depth enhances the distance of the now significant shoot–root oxygen diffusion (Supplementary Figure S5a), which is further exacerbated by the increased initial oxygen consumption burden of a larger root volume (Supplementary Figure S5b). These effects enhanced both the first abrupt and later more gradual decline in root oxygen levels for deeper roots. Following the decline, a

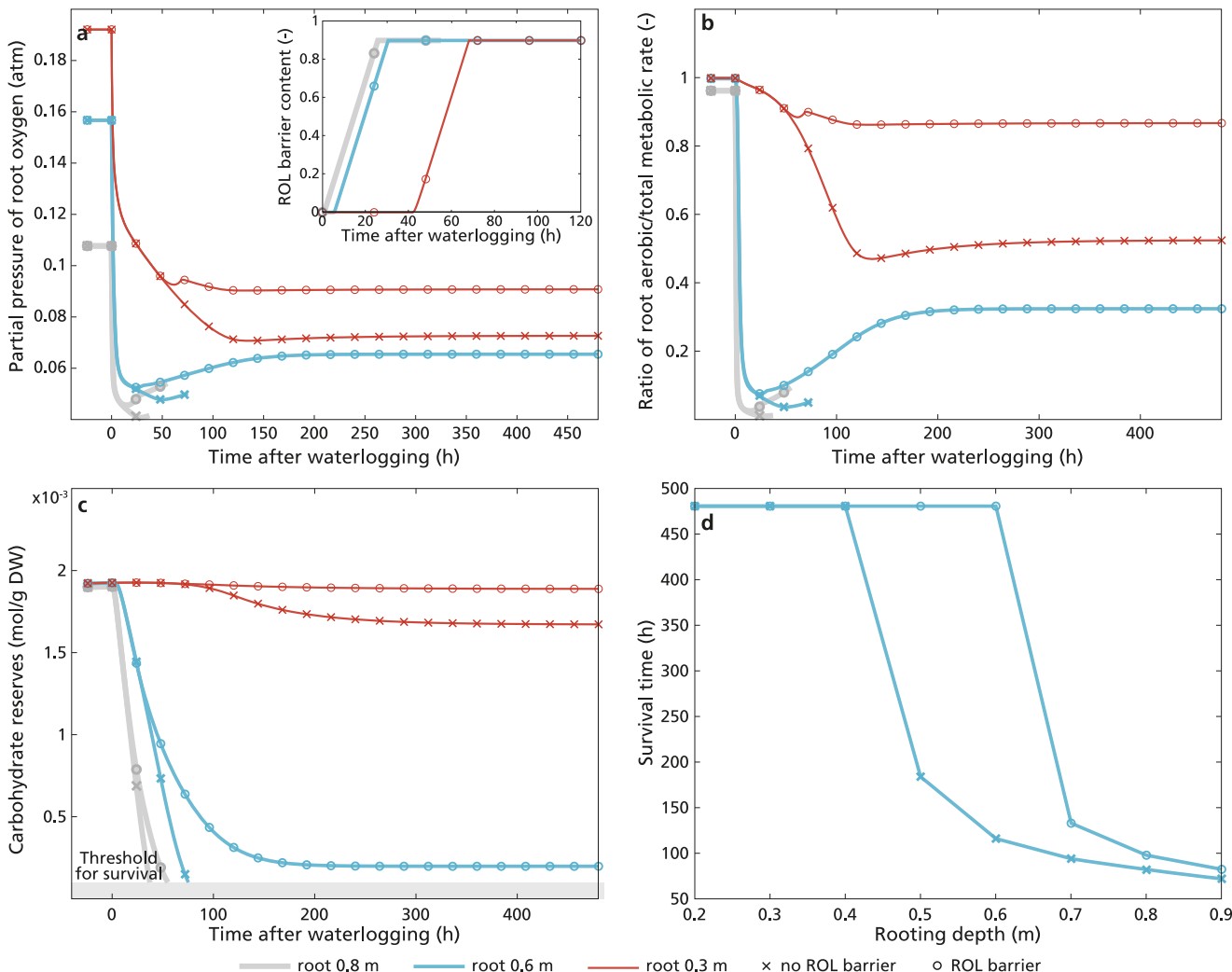

**Figure 4.** Dynamics of (a) ROL barrier induction and root oxygen concentration, (b) the ratio of aerobic metabolic rate in the total metabolic rate, and (c) carbohydrate reserves (plant death occurred when carbohydrate reserves dipped below $10^{-4}$ mol g$^{-1}$ DW, illustrated in grey area) of plants with rooting depths of 0.3 m, 0.6 m and 0.8 m with the presence of an aerenchyma level 0.5 after 20 days (480 hours) upon the initiation of waterlogging. Simulation stops upon plant death. In panel (d) again, we presented the survival duration of plants across a range of rooting depths from 0.2 m to 0.9 m, with the presence of an aerenchyma level of 0.5 during prolonged waterlogging.

minor recovery in root oxygen levels occurs as a result of the ongoing decline in carbohydrate levels, and thus of the remaining fraction of aerobic metabolism (Figure 4b). This recovery, typically occurring around 50–110 hours after the initiation of waterlogging, was further reinforced by and sped up by the formation of ROL barriers (Figure 4a), enabling plants to maintain a larger fraction of the aerenchyma-supplied oxygen and resulted in a partial recovery of aerobic metabolism (Figure 4b and Supplementary Figure S5b) and stabilization or carbohydrate levels (Figure 4c). If roots were too deep and minimum oxygen levels reached were too low, oxygen recovery was insufficient to rescue the plant from carbon starvation (Figure 4a–c, rooting depth 0.8 m). As a consequence, ROLB formation enhanced the window of rooting depths for which survival occurs (Figure 4d).

### 3.6. ROL barrier-mediated survival requires a minimum aerenchyma content

To investigate how the relevance of ROL barriers for waterlogging survival depends on aerenchyma level, we next compared survival of plants with and without ROL barrier formation for varying aerenchyma levels and a constant intermediate rooting depth of

0.6 m. The highest used aerenchyma level of 0.66, corresponding to 66% cross-sectional tissue porosity, is based on the highest found aerenchyma levels in rice, with lower levels corresponding to observations in maize and wheat (Pedersen et al., 2021). We found no substantial effect of aerenchyma content on the timing of ROL barrier formation (Figure 5a, in-set). In contrast, aerenchyma content had a significant effect on root oxygen levels reached during the initial rapid and secondary, more gradual decline phase (Figure 5a), with both higher aerenchyma content and ROL barrier presence reducing this decline. In presence of zero or limited aerenchyma content, root oxygen levels and carbon reserves collapsed before the completion of ROL barrier formation, preventing the ROL barrier mediated partial recovery of oxygen levels (Figure 5a) and aerobic metabolism (Figure 5b) that enables survival of carbohydrate starvation (Figure 5c) that we observed for higher aerenchyma contents. Investigating a broader range of aerenchyma levels, our results indicate that a minimum level of aerenchyma content is essential for ROL barriers to result in waterlogging survival (Figure 5d).

The observed increase in survival window to either larger rooting depths (for constant aerenchyma, content Figure 4d) or lower aerenchyma content (for constant rooting depth Figure 5d)

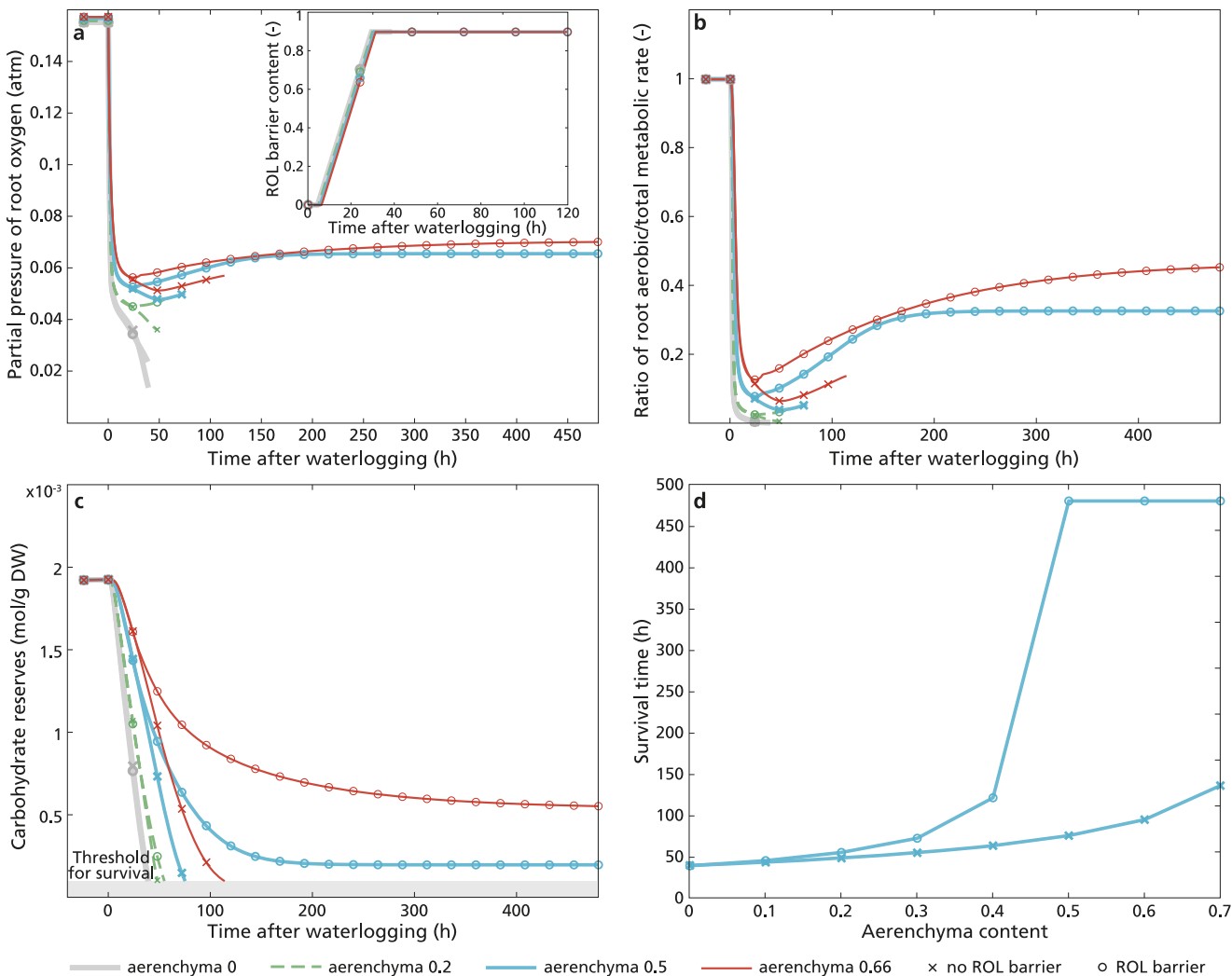

**Figure 5.** Dynamics of (a) ROL barrier induction and root oxygen concentration, (b) the ratio of aerobic metabolic rate in the total metabolic rate, and (c) carbohydrate reserves (plant death occurred when carbohydrate reserves dipped below $10^{-4}$ mol $g^{-1}$ DW, illustrated in grey area) of plants with aerenchyma content 0, 0.2, 0.5 and 0.66, with constant rooting depth of 0.6 m after 20 days (480 hours) upon the initiation of waterlogging. Simulation stops upon plant death. In panel (d), we presented the survival duration of plants across a range of aerenchyma content levels from 0 to 0.7 in plants with a rooting depth of 0.6 m during prolonged waterlogging.

was observed independent of precise parameter choices were made for plant architecture (Supplementary Figure S6), the induction of stomatal closure (Supplementary Figure S7), or the oxygen exchange between shoot in root (Supplementary Figure S8).

### 3.7. Intermediate ROL barrier induction timing optimizes plant survival

The timing of ROL barrier formation depends both on rooting depth and, hence, the rate at which rhizosphere oxygen levels drop below a certain threshold, as we saw in Figure 4, as well as on the oxygen threshold below which ROLB barrier formation is induced. Therefore, we also, investigated the effect of timing of ROL barrier induction on plant survival by varying the threshold rhizosphere oxygen level ($[O_2]'_{rhizo}$, see Supplementary Table S1) below, which ROL barrier induction occurs between 0.18 (early induction), 0.1 (reference value) and 0.01 (late induction). We then ran the model across the previously defined ranges of rooting depths and aerenchyma content levels, with a maximum ROL barrier formation level at 0.9. Our results indicated that intermediate (reference) timing enhances plant survival more effectively than

either early or late induction (Supplementary Figures S9 and S10). This outcome could be explained by the fact that the reference timing allowed ROL barrier formation to complete approximately when root oxygen level began to fall below the rhizosphere oxygen level, and substantial oxygen loss to the rhizosphere would occur. Earlier induction would inhibit rhizosphere–root oxygen transport already when this transport is still providing a net influx to the root, while later induction may come at a time when oxygen and carbohydrate levels have dropped below a point where recovery is possible (for further details see Supplementary Information). Still, differences between early and intermediate timing are relatively minor, and mostly affect survival duration under conditions with no long-term survival, whereas late timing has more pronounced effects and reduces the window of rooting depths and aerenchyma content in which long-term survival may occur.

## 4. Discussion

In this study, we developed a minimal model that aims to encapsulate the morphological and anatomical responses of typical plants to waterlogging. Our model results indicate that reducing rooting

depth, developing aerenchyma and inducing ROL barriers all contribute to improved plant waterlogging survival. While reduced rooting depth and ROL barriers in isolation have limited effect on survival, their effectiveness is significantly increased when combined with the induction of aerenchyma. Importantly, an increase in one factor can, to some extent, compensate for the decrease in one other. Additionally, we find that plants exhibiting high levels of aerenchyma and ROL barrier content can survive prolonged waterlogging even for relatively large rooting depths. This finding aligns with observations in wetland plant species such as rice and *Phragmites australis*, with high aerenchyma content and inducible ROL barriers, and rooting depths generally greater than less flood-tolerant plant species, such as Arabidopsis (Colmer, 2003b; Geng et al., 2023). Our model also shows that ROL barriers enable survival of prolonged waterlogging only when combined with a minimum level of aerenchyma content, with larger rooting depth requiring higher aerenchyma content. This finding coincides with the observation that flood-tolerant land plant species like maize and wheat, which typically have shallower roots and less aerenchyma content, do not induce ROL barriers (Guo et al., 2021; Hanslin et al., 2017; Yamauchi et al., 2014). Our model explains these observations, showing that only under sufficient aerenchyma content, promoting shoot–root oxygen transport is root oxygen is maintained by ROL barriers.

We further observe that model plants with greater rooting depth tend to exhibit an earlier induction of ROL barriers, consistent with experimental findings (Shiono et al., 2011). Additionally, we explored the impact of ROL barrier induction timing. Our results indicate that optimal timing of ROL barrier formation coincides with the period when root oxygen level decreases below ambient soil oxygen level, allowing the ROL barriers to kick in timely to prevent radial oxygen loss while not yet hindering oxygen flow when the oxygen gradient is still oriented towards the root. However, such optimal timing and its implications still await experimental confirmation.

**Open peer review.** To view the open peer review materials for this article, please visit http://doi.org/10.1017/qpb.2025.10016.

## Acknowledgements

We thank A.W. Markus for assistance with graphical design.

**Competing interest.** The authors declare no competing interests.

**Data availability statement.** Model code that was used is freely available at: https://github.com/kirstentt/flooding-stress.git (Zenodo DOI pending).

**Author contributions.** H.d.B. and K.t.T conceived the study, S.C. wrote the simulation code, performed simulations and plotted results. S.C., H.d.B. and K.t.T analysed the generated data and wrote the manuscript.

**Funding statement.** S.C. was supported by a Utrecht University CSF (Complex Systems Fund) grant (no grant number available).

**Supplementary material.** The supplementary material for this article can be found at http://doi.org/10.1017/qpb.2025.10016.

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
