## [Reviewer Report]

The manuscript by Chen et al describes a simple model of responses to waterlogging. The authors develop a minimal two-component model that captures the key physical parameters. Despite the simplistic model, there are some very interesting insights are discussed. The fit to the data seems good. The article is well-written and mostly very clear.

I got a little confused by the modelling details. Based on the equations and the supplemental material, the flows, Q_xx, seem to have different units. For instance, Q_SDO seems to have units of mol m2 s-1 x mol m-3 x m2 = mol2 m s-1 whereas Q_SRO seems to have units of mol m2 s-1 x mol m-3 x m-2 = of mol2 s-1 m-3, but both should be mol s-1. Even using the correct unit for the diffusion constant there still remains a discrepancy.

A key parameter in the simulations and results discussion in root depth. However, the length direct dependence in the model was not obvious to me; it seems it is rather a surface area dependence, which makes sense for gas exchange. Unless I misunderstood, I suggest changing root depth to root surface area and discussing the findings in this context to not confuse readers.

What is n in many of the equations? What is its numerical value?

What is r_LAI in the equation for S_canopy?

In the equation for D_soil, what is H. From the equations it seems impossible that D_soil could be D_water even is no air is present in the soil?

The authors seem to use aperture to describe stomatal conductance. The units, however, don’t seem to match for either.

Diffusivity is another term for diffusion constant. In the supplemental table they have different units.

I failed to find any reference to a diffusion constant or diffusivity in the provided reference, Bailey et al.

Supplemental material: Chemical entities are not part of the units and should be removed.

---

## [Editor Report]

Dear Prof Ten Tusscher,

Thanks for submitting your paper to QPB. Apologies that it’s taken a while to collect reviewer comments. After requests from journal staff I have acted as one reviewer for this article in addition to my role as a handling editor. The editor-in-chief will review this correspondence and can be contacted in case of any conflicts arising.

The review comments mainly focus on the structure of the model, from specific terms and processes (including Hill coefficients, and units balancing) to the level of coarse-graining and motivation for some structural features. If these questions can be answered I think it will help increase the trustworthiness of the model, and convince readers that its structure is better than alternatives. There are also some other points about the presentation and interpretation of the results (and code).

We’d be delighted to consider a revised manuscript that addresses these points. I’m also available to discuss the points that have been raised through the review process if this would be helpful.

All the best,

Iain Johnston

--- Reviewer 2 comments

The article builds a compartmental, ODE-algebraic model for metabolite transport and reactions, focussing on plant survival in waterlogging (and hence hypoxic) conditions. The constructed model is interrogated to explore the influence of three key control parameters: rooting depth, aerenchyma content, and ROL barrier presence. The results are presented as a scan through the space of these parameters and a more detailed look at the dynamics of the system in some specific instances.

The paper is a new approach to an interesting problem. I have several questions about the model construction (and presentation), and some suggestions for how the results could be interpreted more generally. I do suspect that making the paper as accessible and transparent as possible may require some large-scale (though conceptually simple) changes, so am recommending majors.

My biggest comments are:

1. To me the more natural ordering to present the results would be the parameter scan first (Fig. 5) then the more detailed view of the dynamics in specific cases later (Figs. 2-4). Then the reader gets an overview of the main behaviours first, then more detail later.

2. All the survival time behaviour in Fig. 5 shows pretty dramatic switch-like behaviour -- low survival time (blue), fast transitioning to high survival time (yellow). We see this as well in Fig 3d, 4d. But how much is this rapid switch a function of the magnitude of the (mysterious) nonlinearity in the equations of motion (i.e. the Hill coefficients n)?

3. The model is described as a “minimal model”. What specifically is minimal about it? To my eyes it sits at a slightly awkward intermediate point between totally bottom-up (where we rely on our ability to capture low-level behaviour and assume that the emergent behaviour must then be accurate) and totally top-down (where we are guided by the shape of data and select a model trading off over- and under-fitting). Lots of model elements have a bottom-up feel, but then several important processes are assigned a heuristic model term. More broadly -- what if we linearised some terms? Removed some constants? Put more details into some parts of the model? Would we do better?

4. l113 “we used plant root ATP status as a proxy to control stomatal aperture” -- this (key) structural feature stands in stark contrast to the mechanistic detail of the other models. It feels awkward -- (a) after lots of detail on transport between compartments, ATP teleports from the root to the leaf and (b) after careful consideration of several influences on other processes, ATP is the sole determinant of stomatal behaviour. Can these simplifications be justified, e.g. with reference to literature?

5. Please label the equations throughout the model. It would help a lot if specific equations were referenced throughout the results -- for example when the control parameters are varied, the corresponding equation(s) containing those parameters could be linked to. 

6. Please unpack the zip file in the Github repo! 

More specific, smaller comments below. Most of these would at most need a single-sentence answer or small change; the ones marked * may be a bit more involved.

Abstract -- anoxia or hypoxia?

Abstract could be more accessible -- not sure aerenchyma and ROL are generally understood? Could they have a half-sentence introduction?

l 30 shoot-root ratio -- of what?

l 51 thus? what is the logical connection implied here?

l 69 subtitle feels incomplete. assumptions in?

Fig 1 / l 76 -- stem and root have same diameter, but are given different symbols (R_p, R_r)?

l84 not sure of some of the geometric assumptions here. why is R_rhizo = 2 R_r? why have two different symbols? why R_bulk = 4 R_r? two different symbols?

confusing to have “rooting depth” without a symbol in these expressions

l86 if rhizosphere has radius 2 R_r then how can it have width R_r ? do you mean the “additional” width?

l 86 Z_r seems to be the “rooting depth” I just asked for but it isn’t in Fig 1

l95-96 please label all equations!

* so -- although we have a spatially-embedded model we don’t have spatial dependence in the metabolites and their concentration profiles? why does the cylindrical geometry matter?

l118 and throughout, is (CH2O)6 a standard way of writing glucose? to me it sort of implies six identical monomers.

l128, 139, 140, 148, etc -- are all these Hill coefficients the same? what are they? why? (see main point 2)

Throughout equations -- I’m not convinced the units in the expressions always balance?

* perhaps it could be made clearer that the control parameters we’re fundamentally varying are rooting depth, aerenchyma, and ROL barrier.

Fig 2 / line 226 -- not really a plateau!

l 279 -- do the results from varying root depth control for just the total amount of root?

l283 -- “causes enables”

I think it would help me follow if the results subsection titles were themselves results with a direction, e.g. “Less deep rooting enhances waterlogging survival”

l290 section -- what model component is specifically turned on here? which equation? l292-293 could be explained a lot more (see main point 5)

l307 typo occursa

l324-325 -- significant usually refers to statistics. substantial?

l359 fig 8 is supp fig 8

l 379 -- no need for italics in Supplementary

---

## [Reviewer Report]

The revised manuscript is much improved.

I have only minor comments:

I would advise against using the same symbols to denote different physical entities (the various Q values do not have the same units, eg line 205 and 206).

I would advise against using the symbol ‘o’ as an exponent (line 178).

I suggest placing the exponents (superscripts) in the equations directly after the term of interest and not after the subscript, eg [O<sub>2</sub>]<sup>r</sup><sub>rhizo</sub> instead of [O<sub>2</sub>]<sub>rhizo</sub><sup>r</sup>. This affects most equations.

There are discrepancies with how units are written, eg sometimes m^3 and sometimes m<sup>3</sup>.

There are several missing spaces after mathematical terms.

---

## [Editor Report]

Thanks very much for your careful consideration of the reviewer comments. To my eyes the restructuring of the paper and inclusion of additional quantitative information helps tell the story more clearly, and have improved the transparency and interpretability of the results. The codebase is now more accessible (perhaps some reformatting of the README would make things clearer? It looks like text-file whitespace, which doesn’t translate into Markdown, has been used). I am happy to recommend acceptance for this work, which provides new scientific insight using a quantitative modelling approach.